# Effect of Different Amine Catalysts on the Thermomechanical and Cytotoxic Properties of ‘Visco’-Type Polyurethane Foam for Biomedical Applications

**DOI:** 10.3390/ma16041527

**Published:** 2023-02-11

**Authors:** Dominik Grzęda, Grzegorz Węgrzyk, Adriana Nowak, Gabriela Komorowska, Leonard Szczepkowski, Joanna Ryszkowska

**Affiliations:** 1Faculty of Materials Science and Engineering, Warsaw University of Technology, Wołoska 141, 02-507 Warsaw, Poland; 2Department of Environmental Biotechnology, Lodz University of Technology, Wolczanska 171/173, 90-530 Lodz, Poland; 3Fampur Adam Przekurat, 83-305 Bydgoszcz, Poland

**Keywords:** viscoelasticity, elastic polyurethane foams, catalysts, cytotoxicity, biomedical applications

## Abstract

Components for manufacturing polyurethane foams can adversely affect the human body, particularly if they are in contact with it for long periods. In applications where the foam is not placed directly into the body, the study of the product’s effects is often neglected. In the case of human skin, distinguishing the increasingly frequent problems of skin atopy, more attention should be paid to this. This paper presents the influence of the different catalytic systems on cytotoxic and thermomechanical properties in polyurethane foams. Among others, foams were produced with the most popular catalysts on the market, DABCO and a metal-organic tin catalyst. The foams were characterized by thermomechanical properties and were subjected to a cytotoxicity test against human keratinocytes. In biocompatibility tests with skin cells, the results were highly variable. VAB 2 with a catalytic system consisting of commercial Diethanolamine and Addocat^®^105 performed the best. However, with such a catalytic system, the mechanical properties have worsened.

## 1. Introduction

Polyurethanes are used in many fields, such as, e.g., the building industry [1], automotive [2], furniture [3], coating [4], and earthquake engineering [5,6]. They can also be used in the medical industry, as their properties are extremely dependent on the components from which polyurethanes are formed. In recent years, much emphasis has been placed on developing new materials in medical-related fields. The key aspect is biocompatibility with human tissue. Depending on the intended use and the duration of its contact with the organism, the materials are subject to different biocompatibility testing protocols.

Regarding skin contact, an initial cytotoxicity assessment of the material and its effect on sensitivity and skin irritation is performed [7]. Nowadays, the most common prosthetic liners are made from silicone rubber. Although it is believed to be biologically inert, there are increasing reports of contact allergy to silicone from medical devices, including ventriculoperitoneal valves, breast prostheses, and cosmetic fillers [8,9]. In the case of polyurethane foams, they are formed when polyols, isocyanates, catalysts, surfactants, and foaming agents are mixed. It has been proved that exposure to isocyanates can cause asthma and contact dermatitis [10]. There are many polyurethane catalysts, but the foaming and curing of polyurethanes are mainly based on amines [11]. Their mechanisms of action are very well described in the article by Silva and Bordado [12]. Primary aliphatic amines have been shown to affect the skin adversely and are highly irritating mucous membranes [13]. Tertiary amines have proven useful as catalysts in the production of polyurethane foams. They accelerate blowing reactions and urethane formation. However, numerous tertiary amines have a strong and fishy odor that is transferred to the polyurethane foam [14]. Some of these amines also exhibit toxicity. In skin contact with amine catalysts, there is a risk of severe irritation and even burns. Repeated or prolonged exposure can also cause severe contact dermatitis.

In most cases, this will result in irritant dermatitis [15]. One solution to odor and toxicity problems is to use tertiary amines containing a reactive group that will bind to the isocyanate and forms a hard chain segment of the polyurethane foam. This way, the diffusion of the amine out of the foam is prevented. It minimizes the odor and possible health risks associated with the free amine and eliminates potential sensitization reactions caused by the isocyanate.

Polyurethane catalysts can be broadly divided into organic acidic, organic basic, and tin-based catalysts. Organic basic catalysts activate the OH group via a nucleophilic mechanism, while organic acid catalysts activate the NCO isocyanate group via an electrophilic mechanism. The popularity of organic catalysts is steadily increasing due to the trend toward reducing the environmental impact that tin undoubtedly causes [16]. The influence of the type of catalysts on the characteristics of polyurethane foams and, in particular, their cytotoxicity is rarely discussed in the literature. A description of the influence of the type of substrate used is contained, among others, in the work by Krebs and Hubel [17]. 

This work proposes using different systems of reactive catalysts to produce viscoelastic polyurethane foams. They will be used as components of orthoses and liners in contact with the skin. The study aims to characterize the foams produced with these catalysts. Due to stringent requirements, the polyurethane foams used inside the human body are tested for cytotoxicity against human cells [18,19]. The cytotoxicity study is often disregarded in external applications. This research aimed to prove the validity of such studies in applications involving skin contact (such as leg or arm prostheses, orthoses, and mattresses) and investigate the effect of the most industrially popular tin catalyst based on the toxic heavy metal, tin.

## 2. Materials and Methods

### 2.1. Materials

In this study, conventional polyurethane foams were synthesized. Polyols, which were used to manufacture the foams, are Voranol CP 1421, a high ethylene oxide content polyether triol (33 mg KOH/g, AMw 5023 g/mol, Dow, Midland, MI, USA), Rokopol D2002—polyoxypropylenediol (53–59 mg KOH/g, AMw 2000 g/mol PCC Rokita) and Rokopol G500—polyoxyalkylenetriol (290–310 mg KOH/g AMw ~560 g/mol, PCC Rokita).

The isocyanate used was Ongronat TR 4040—a mixture of MDI (methylene diphenyl diisocyanate) mixed isomers and oligomeric MDI (31.6–33.6%NCO/wt.%, BorsodChem, Hungary). The silicone surfactant was TEGOSTAB^®^ B404 (Evonik, Essen, Germany). 

In addition, the catalysts described below were used:DABCO NE 1070—*N*,*N*-dimethyl aminopropyl urea,(H_2_NCO)NH(CH_2_)_3_N(CH_3_)_2_, molecular weight 145 g/mol (Air Products); trifunctional reactive amine catalyst;JEFFCAT^®^ MDPTA (Huntsman, Hungary) dimethyldipropylenetriamine; trifunctional reactive amine catalyst;JEFFCAT^®^ DPA (Huntsman, Hungary)—*N*-(3-dimethylamino-propyl)-*N*,*N*-diisopropanolamine; reactive amine(III-terminal amine) catalyst with two hydroxyl groups;Addocat^®^105 (Rhein Chemie) is a tri-ethylene di-amine—DABCO solution in dipropylene glycol;Diethanolamine (PCC Rokita)—pH adjuster, diethanolamine 89 ÷ 91%; water, 9 ÷ 11, DELA;Niax A-1—70% bis(2-dimethylamino ethyl)ether and 30% dipropylene glycol;DBTDL—dibutyltin dilaurate, tin catalyst.

Component A, polyols Voranol CP 1421, Rokopol D2002, and Rokopol G500 were used to make the foams, with the proportion of each polyol being the know-how of the article’s authors. Ongronat TR 4040 was used as component B. Foams were produced with an isocyanate index (INCO) of 0.9. The function of the porophore was performed by distilled water at 3 php (parts according to one hundred parts of polyol, wt/wt).

Polyurethane foams were made using different catalysts, as described in Table 1. Their quantitative contribution is the authors’ know-how.

### 2.2. Methods

#### 2.2.1. Synthesis Parameters and Apparent Density

A 1 s precision electronic stopwatch was used to determine the start time (the beginning of foam formation after mixing of components A and B), the rise time (up to the highest height of the foam), and the gel time (as long as the viscosity of the mixture is sufficient to pull a string out of a polymer with a rod).

According to EN ISO 845 [20], the sample’s mass and volume were measured in order to determine the apparent density. Using a WPA 180/C/1 (Radwag, Radom, Poland) analytical balance, the weight of the samples was calculated to the nearest ± 0.1 mg. The cubes with dimensions of 50 × 50 × 50 mm^3^ were cut from the samples and measured to the nearest ±0.1 mm.

#### 2.2.2. Fourier Transform Infrared Spectroscopy

By using absorption spectra from a Nicolet 6700 spectrophotometer (Thermo Electron Corporation, Waltham, MA, USA) combined with an ATR module, the chemical structure of the foams was examined. Each sample has been scanned 64 times in the 4000–400 cm^−1^ wavelength range. Absorbance units were used to measure the amount of IR radiation absorbed by a sample.

The chemical composition of the foams was analyzed using absorption spectra obtained with a Nicolet 6700 spectrophotometer (Thermo Electron Corporation, USA) equipped with an ATR module. Each sample was scanned 64 times in the wavelength range of 4000–400 cm^−1^. Absorbance units were used to measure the amount of IR radiation absorbed by a sample. Software called Omnic Spectra 8.2.0 was used to analyze the results (Thermo Fisher Scientific Inc., Waltham, MA, USA). A representation of three spectra was made for each conventional VAB foam. 

#### 2.2.3. Scanning Electron Microscope Observations

The surface of the foams was examined using a scanning electron microscope (SEM) (HITACHI SU8000, HITACHI, Tokyo, Japan). The samples were sputtered with gold, and the layer was 10 nm. The resulting images were merged into a larger-area panorama using a graphic program. The panorama provided an opportunity to increase the area of pore size analysis and thus introduce better statistics of larger pores that did not fit into the working area. Once the panoramas were made, each pore was manually outlined using a graphics tablet to obtain the binary image needed for the pore size analysis. The image thus prepared was analyzed using MicroMeter 1.0. (Figure 1)

#### 2.2.4. Thermogravimetric Analysis

A TGA Q500 (TA Instruments Waters, New Castle, DE, USA) was used for thermogravimetric analysis. Samples were heated at a rate of 10 °C/min in an air environment from room temperature to 900 °C. The collected data were analyzed using the Universal Analysis 2000 ver.4.5A program (TA Instruments Waters, USA).

#### 2.2.5. Rebound Resilience and Elastic Recovery Time

The resilience to rebound was assessed in accordance with EN ISO 8307 [21]. On a 10 × 10 × 10 cm^3^ sample cut from the inside of a foam element, a 1.6 cm steel ball was dropped from a 50 cm height. Slow-motion video analysis was used to measure the rebound’s height.

After releasing a 10 × 10 × 10 cm^3^ sample that had been compressed at 90% for 1 min at room temperature, the elastic recovery time was measured. An electronic stopwatch was used to measure the time to the nearest 1 s [22]. 

#### 2.2.6. Compression Stress Value and Compression Set

Compression tests were performed according to EN ISO 3386 [23] on an Instron 5565 double-column test machine. The samples were compressed by 70% of their initial height. Three initial loading and relaxation cycles were performed, and the signals from the fourth compression cycle were collected. From the compression test, the hardness of the foam (stresses at 40% of the sample height) was determined. Stress at 25% and 65% of the sample strain was also collected from the fourth loading cycle to determine the SAG comfort factor by dividing the stress of the sample at 65% by stress at 25%. This allows the evaluation of cushioning quality. A high value indicates resistance to bottoming out as the load is applied. 

According to EN ISO 1856 [24], the compression set was marked. The samples with dimensions 50 × 50 × 25 mm^3^ were compressed for 50% and 90% of the initial height at 70 °C for 22 h in a parallel direction to the foam rise. After removal of the compression and waiting 0.5 h, the height of the samples was remeasured to assess if there was any dimension change. The permanent deformation was calculated by percentage loss in the height of the specimens.

#### 2.2.7. Dynamic Mechanical Analysis

DMA Q800 machine (TA Instruments Waters, New Castle, DE, USA) was used to perform dynamic mechanical analysis in multi-frequency strain with a compression clamp size of 40 mm. Cylindrical samples with a diameter of 12.8 mm and height below 10 mm were measured according to parameters: frequency 3 Hz, oscillation amplitude 30 µm, static force 0.01 N, force track 125%, minimal oscillation force 0.00001 N. The samples were cooled to −90 °C and kept at that temperature for 5 min, then with ramp 3 °C/min heated to 120 °C. The results were analyzed using Universal Analysis 2000 ver.4.5A software (TA Instruments Waters, USA).

#### 2.2.8. Sweat Absorption Test

For each foam variant, the measurements of artificial sweat absorption in both acidic and alkaline pH were taken. According to ISO 105-E04:2013 [25], sweat solutions were prepared. Samples were cut into cubes of a total of 1 g weight, then dried under vacuum at 70 °C for 24 h, and weighed again using WPA 180/C/1 (Radwag, Poland) analytical balance to assess the dry weight (W_d_). In the next step, samples were soaked separately in sweat solutions for 24 h. Before weighing to determine properly the equilibrium swelling weight (W_s_), the excess liquid was removed from the external sample for ten minutes on new filter paper made by Thermo Fisher Scientific in the United States. The equilibrium swelling mass was divided by the dry foam mass to determine the equilibrium swelling index (Q).

#### 2.2.9. Foam Extracts Preparation

The extracts were prepared in accordance with EN ISO 10993-12:2012 [26] for cytotoxicity testing. Each foam was weighed (2.0 g ± 0.05 g), mixed, and extracted for 8 h at 37 ± 1 °C (300 rpm) in 40 mL of complete culture Dulbecco’s Modified Eagle’s Medium (DMEM) (Merck Life Science, Warsaw, Poland). Syringe filters of 0.22 µm and 0.45 µm were used to sterilize each extract (Labindex S.A., Warsaw, Poland).

#### 2.2.10. HaCaT Cell Culture 

The conventional immortalized human keratinocyte cell line HaCaT (the initial material developed by Prof. Dr. Petra Boukamp and Norbert Fusenig [27]) was utilized in the study. Cells were bought from Cell Line Service GmbH (Eppelheim, Germany) from the 35th passage. They were grown as a monolayer in DMEM with 10% fetal bovine serum (FBS), 2 mM gluta-MAX^TM^ (Gibco, Thermo Fisher Scientific, USA), 25 mM HEPES (Merck Life Science, Warsaw, Poland), 100 g/mL streptomycin, and 100 IU/mL penicillin mixture (Merck Life Science, Warsaw, Poland) in a Galaxy 48S incubator (New Brunswick, United Kingdom) for 3–5 days in 5% CO_2_ atmosphere at 37 °C. According to the manufacturer’s instructions, cells were detached using TrypLETM Express (Gibco, Thermo Fisher Scientific, USA), centrifuged (182× *g*, 3 min), decanted, and fresh DMEM was added after reaching 80% confluence. An exclusion test using trypan blue (Merck Life Science, Warsaw, Poland) was used to determine viability. The viability of cells taken to each experiment was 90–95%.

#### 2.2.11. Neutral Red Uptake (NRU) Assay 

The NRU assay was used to assess cytotoxicity in accordance with ISO 10993-5 [28]. The cells were seeded at 10,000/well in a full culture medium into clear flat-bottom 96-well plates and incubated at 37 °C for 24 h in 5% CO_2_. The following concentrations of the test extracts were added to the culture medium in four replicates per experiment: 1.56%, 3.125%, 6.25%, 12.5%, 25%, 50%, and 100%. Three independent experiments were conducted. A vehicle was cells in a culture medium; a positive control was cells incubated with dimethylsulfoxide (DMSO, Merck Life Science, Warsaw, Poland) at concentrations from 0.156% to 10%. The assay was conducted for 24 h. After that, the tested samples were aspirated, NR (Merck Life Science, Warsaw, Poland) was added to phosphate-buffered saline (PBS, Merck Life Science, Warsaw, Poland) at a concentration of 50 g/mL, and the plates were incubated for an additional 3 h. The NR solution was gently aspirated from the cells and removed using a freshly produced desorbing solution (1% acetic acid, 50% ethanol, and 49% distilled water). The absorbance was measured at 550 nm using a 620 nm reference filter in a TriStar2 LB 942 microplate reader (Berthold Technologies GmbH and Co. KG, Bad Wildbad, Germany). The percentage of cell viability was estimated by comparing the sample’s mean absorbance to the control’s mean absorbance. The obtained curves were used to calculate IC50 values and non-toxic/safe doses/concentrations (IC0) of polyurethane foam extracts. Microscopically, morphological differences in HaCaT cells were assessed using an inverted Nikon Ts2 with EMBOSS contrast (Nikon, Tokyo, Japan) and the Jenoptic Subra Full HD Color at a total magnification of 100. A qualitative microscopic study was performed, with the grade ranging from 0 to 4 based on the responsiveness of the cells to the substances, where 0 indicates no reactivity and 4 indicates strong reactivity.

## 3. Results

### 3.1. Synthesis Parameters and Apparent Density

Different catalyst systems alter the synthesis process’s course, changing the start time, growth time, and gelation time. The results of the synthesis parameters measurements are put in Table 2.

Start time gives information about how long the molds can be filled before foam growth starts. Rise and gel times determine the time required for the molds’ circulation, affecting the products’ price.

The change in the arrangement of catalysts in the manufactured foams changes the synthesis course. Start, rise, and gel times for VAB 3 foam were accordingly 2 s, 60 s, and 90 s between 2 and 17 s. For VAB 1 and VAB 2, the start time was 15 and 17s, the rise time was 420 and 255 s, and the gel time was 560 and 300, accordingly. The range of foaming parameters is wide, allowing products of different shapes to be formed [22]. The shortest synthesis parameters characterized the foam with DBTDL. The relative activity of organometallic catalysts, including DBTDL, is significantly higher than that of tertiary amine catalysts [29]. The main task of DBTDL is to grow the polymer chain efficiently enough. The change in the catalyst system results in slight changes in the apparent density of the foams, with the highest density for VAB 1 foam, which also has the longest gelation time.

### 3.2. Fourier Transform Infrared Spectroscopy

The obtained spectra of the samples were merged in Figure 2.

The FTIR spectra of the analyzed foams are similar. In the FTIR spectra of the foams, bands in the range of 3600–3400 cm^−1^ come from stretching vibrations of the −OH groups of the polyol hydroxyl groups. Their occurrence is related to the use of INCO=0.9 in the synthesis process, i.e., there are fewer isocyanate groups in the reaction environment compared to the other hydroxyl OH and amino NH_2_ functional groups. Therefore, a broad peak is observed in the spectra in the 3400–3200 cm^−1^ range. It results from asymmetrical and symmetrical stretching vibrations of the amino −NH_2_ group present in the urethane groups, urea derivatives, and/or the rest of the catalysts. The bands with maxima at wavenumber values of 1540 cm^−1^ and 1513 cm^−1^ show that the amino −NH2 group is deformed. The bands at 2975 cm^−1^ and 2872 cm^−1^ are caused by asymmetrical and symmetrical stretching vibrations of the carbon-hydrogen −CH group from the CH3 and CH2 groups. These groups also produce a band with a maximum of 1453 cm^−1^ (CH3) and 1373 cm^−1^ (CH2) due to asymmetrical and symmetrical deformation angles, as well as a band with a maximum of 1306 cm^−1^ due to stretching vibrations. The existence of carbonyl C=O groups in the urethane and urea groups is confirmed by multiplet signals in the 1770–1630 cm^−1^ range. In the aromatic ring, the stretching vibrations of the C=C group are the source of the signal values of 1597 cm^−1^, while the stretching vibrations of the C−N group are the source of the band for the wavenumber of 1227 cm^−1^. The C−O group stretching vibrations that result in the formation of elastic polyurethane segments are what give rise to the signal from the maximum at the wavenumber of 1088 cm^−1^.

Figure 3 shows that the foams have distinct multiplet signal shapes in the FTIR wavenumber range of 1760–1630 cm^−1^. The stretching vibrations of the carbonyl groups in urethane and urea groups, bound with a hydrogen bond or not bound, caused the changes in the intensity of the components that led to the differences [30].

Quantitatively, changes in multiple band components were analyzed, and the procedure for multiple band analysis is described in the paper [31]. Based on the multiple band component analysis results, the degree of phase separation (DPS) and the proportion of urethane groupings (part of urethane groups) in the hard phase were calculated. In addition, the proportion of urethane groupings bound by hydrogen bonds was calculated as the ratio of the peak area derived from the vibration of the hydrogen-bonded urethane groups to the peak area derived from the urethane groups. 

The results of this analysis are summarized in Table 3.

These results suggest that phase separation occurs more readily in the VAB 3 foam, which has the lowest proportion of urethane bonds in its hard segments. The catalyst system used in this foam favors the formation of urethane bonds.

### 3.3. Scanning Electron Microscope Observations

Figure 4 shows images of the tested samples in two directions, perpendicular and parallel to the foam growth.

From the SEM images, it can be concluded that the choice of catalyst system had a significant effect on both the pore structure and the pore size distribution. The samples show a mixed pore morphology. Much smaller openings in the walls connect larger closed pores. In VAB 1 and VAB 2 foams, pores with fairly regular shapes are present, unlike the pores of VAB 3 foam. In VAB 3, the cell windows are very irregular, with shapes observed resulting from rapid disruption of the pore walls. The disruption is probably due to a tin catalyst, which made the polymerization reaction very fast. The rapid increase in carbon dioxide formed caused increased pressure in pores. As a result, pores walls were losing continuity. These observations correlate with the foam synthesis parameters. The take-off, growth, and gelation times were the shortest for the VAB3 foam. Based on SEM images, pore diameters and the coefficient of variation were measured for the foam surface in two directions of foam growth. 

The results of this analysis are shown in Table 4.

On the perpendicular cross-section the largest pores and on the parallel cross-section, the smallest pores were observed for the VAB 3 foam, which had the shortest characteristic times. It follows that the large heterogeneity of the pore shape is related to the fast times of chemical reactions. The most regular shape and the smallest pore distribution range are characterized by VAB 2 foam, which had average values of reaction times compared to other tested samples. It can be concluded that the reaction time has a significant effect on the shape and homogeneity of the pores, but there is no linear correlation. 

### 3.4. Thermogravimetric Analysis

The structure of the foam segments can be inferred from the thermal degradation process using TGA. The course of this process is illustrated by the derived mass change (DTG) curves in Figure 5. The arrows indicate the axis for the curves they refer to. The results are summarized in Table 5.

Degradation of polyurethane foam starts with breaking urethane and urea bonds in a temperature range of 230–290 °C (Stage 1). The hard segments of the foam are falling apart. Stage 2 at 290–350 °C represents the start of the degradation of ester bonds from polyol and the release of isocyanate monomers. The minimum of stage 2 at 350 °C shows the end of hard segments breakage. Stage 3 represents ether bond decomposition from polyols. Stage 4 is the decay of material residues into volatile products, particularly aromatic elements. 

The first stage of the process has the maximum degradation rate in the temperature range of about 271 °C. There is a difference in the maximum degradation rate. The degradation rate of VAB 1 foam is the lowest, reaching 0.14%/°C. The highest maximum degradation rate was observed for VAB 3, which was 0.23%/°C. VAB 3 foam was synthesized with the addition of a tin catalyst; there is a possibility that a synergistic effect of amine and tin catalysts has occurred. Strachota et al. [32] observed a synergistic effect due to the action of the gelation catalyst and the tin catalyst on the urethane bond formation process, which resulted in the formation of more urethane bonds. The mass loss in that stage is 13.2% for VAB 1 foam, and for VAB 3 with tin catalyst, it reaches 13.9%, slightly higher than for VAB 2, in which mass loss was 13.4%. Considering the error in determining mass loss in this stage, it can be suspected that the mass change is similar. The observed differences in the degradation pattern in this step may be due to differences in the proportion of urea and urethane groupings in the rigid segments and/or may be due to a low degree of phase separation. At the second stage of decomposition, the maximum degradation rate was the highest for VAB 2 foam, reaching 0.53%/°C, slightly higher than for VAB 3 foam, which was 0.50%/°C. The maximum degradation rate for VAB 1 remains the lowest, reaching 0.44%/°C. In the third degradation stage, the maximum degradation rate was reached in the VAB 2 sample. The highest degradation rate of the soft phase (stage 2) may indicate that the largest number of hard domains composed of rigid segments are dispersed in this phase, whose presence accelerates the thermal degradation process of this phase. The degradation rate is highest at this stage. In the first two degradation stages, about 35% of the mass loss is observed; in the third stage, about 50%, and in the fourth stage, about 4.5%. The degradation rate in this stage is the lowest. 

### 3.5. Rebound Resilience and Elastic Recovery Time

The characterization of the performance characteristics of the foams began with the assessment of elastic rebound and elastic recovery time. The results of these analyses are summarized in Table 6.

The total deformation in viscoelastic foams results from elastic, flexible, and plastic deformation. The typical elastic deformation range is up to 20%. In the case of the foams produced, this range between 7% and 12%, depending on the catalysts used. 

Those foams exhibit the pneumatic viscoelastic effect. The effect arises from the structure of closed cells with perforated cell membranes. The least elastic rebound characterized the VAB 2 foam, with the lowest degree of phase separation and the highest proportion of urethane bonds observed in the hard phase of this foam. Elastic deformation in the foams is favored by the formation of more urethane bonds in the foams. 

The elastic recovery time of the foams to 90% of their initial height is similar. It is approximately 6 s, and the observed differences result from the measurement’s sensitivity to the speed of the experimenter’s response. Longer return times would indicate a small number of effectively elastic chains [33].

### 3.6. Compression Stress Value and Compression Set

The results of compression tests are shown in Table 7.

None of the foams showed significant permanent deformation when compressed by 50% and 90% at 70 °C. The VAB 3 foam has the lowest hardness, which may be due to the pore structure of this foam. The foams analyzed have a SAG comfort factor of 2.4–3.2. The higher the value of the comfort factor, the higher the comfort of the foam user. Conventional flexible polyurethane foams show SAG comfort coefficients of about 1.9, while flexible foams (i.e., High Resilience) show about 2.9. Literature review shows SAG comfort factors at 2.5–3 for the viscoelastic type of foams [34,35]. Values above 3 in case of long-term use as part of a prosthesis, the patient often wears the prosthesis for most of the day, are very satisfactory. In the study by Liu et al. [36], the effects of mattresses with SAG comfort coefficients of 3.5 and 2 were compared. For the mattress with the higher SAG comfort factor, the load damage for the patients was more than six times lower. In addition, this type of damage occurred on average 4 days later with the mattress with the higher SAG comfort factor.

### 3.7. Dynamic Mechanical Analysis

In the case of visco or shape memory foams, it is important to characterize the changes in strength properties as a function of temperature, which is why an analysis was carried out using DMA. The results are summarized in Figure 6 and Table 8. 

The glass transition temperature of the soft phase determined from the tan delta peak varies with the change in the catalytic system. The tangent of the delta is the ratio of the loss modulus over the storage modulus and hence indicates the tendency of a material to dissipate energy (i.e., more viscous-like) or to store energy (i.e., more elastic-like). VAB 2Energy dissipation capacity is greatest for foam. The highest T_g_ is characterized by the VAB 2 foam, with the smallest degree of phase separation and the highest proportion of urethane bonds. 

Regarding elastic state (below T_g_), the highest elastic modulus E’ at −75 °C is characterized by VAB 1 foam and is 8.8 MPa. For this foam, the highest apparent density is observed. Kang et al. [37] associate an increase in density with an increase in the level of intermolecular interactions and, in the case of PUR foams, with the contribution of hydrogen bonds linking urethane groupings. The index of hydrogen bonds that link the urethane groupings is the highest for VAB 1 (Table 3). The catalytic system in VAB 1 favors the formation of hydrogen bonds between the urethane groupings.

In the plateau region’s highly elastic state (above T_g_), the modulus of elasticity changes with a change in the catalytic system. VAB 3 foam has the highest modulus of elasticity at 100 °C and is 6 kPa. This indicates that the catalytic system used in VAB 3 foam resulted in a structure with the highest crosslinking density according to the ideal rubber theory [38]. The catalytic system in this foam promotes crosslinking by trifunctional polyols and may also promote the formation of biuret bonds between free isocyanate and urea groups [37].

Due to the potential application in contact with the human body, properties at 36 °C were also analyzed. At this temperature, the VAB 1 foam has the highest behavioral modulus E’, and the VAB 3 foam shows the highest damping capacity.

These results suggest that phase separation occurs more readily in the VAB 3 foam, which has the lowest proportion of urethane bonds in its hard segments. The catalyst system used in this foam favors the formation of urethane bonds.

### 3.8. Sweat Absorption Test

Patients using prostheses complain of discomfort caused by the sweat of the limb stump in contact with the liner. Therefore, the use of sweat-absorbing liners is beneficial. The study analyzed the sweat absorbency of the liners before and after contact with sweat.

The human skin’s pH value depends on several factors; the normal value is around 4–6 [39]. Factors that alter this value include genetics, age, sweat, sebum, and various skin diseases. According to research, washing hands with traditional soap increases the pH of the skin in hands by 3 units, an effect that lasts 90 min [40]. Such behavior, repeated daily, adversely affects the repair processes of the epidermal barrier. Skin diseases are mistakenly associated with poor hygiene, with sufferers treating their skin with overly strong cosmetic products. On the other hand, an alkaline pH promotes the growth of batteries, increasing the probability of skin infection [41]. As the skin and, therefore, its sweat have different pH values, the effect of pH on the change in sweat absorption by foams was analyzed. The results of the measurements are summarized in Table 9.

Differences in sweat absorbency may be due to differences in reaction and the different surface tensions of the liquid, investigated in our previous paper [42]. Alkaline sweat has a higher surface tension, so it adheres better to the pore walls.

VAB 3 foam had the highest absorbency of both sweat reactions, probably due to its morphological structure. The cellular windows of the pores of this foam are in the form of differently shaped cracks and elongated ovals; this is more conducive to holding sweat than open oval cellular windows. Given the increased intensity of bacterial growth in alkaline reactions, the greater absorbability of alkaline sweat is more beneficial.

### 3.9. Cytotoxic Activity of Polyurethane Foam Extracts

For a period of 24 h, HaCaT keratinocyte cells were subjected to water extracts of polyurethane foam samples at seven distinct concentrations (ranging from 1.56% to 100%) in four replicates of each concentration. The cytotoxicity-concentration dependence curves for each extract are shown in Figure 7 as the mean ± standard deviation of the mean (S.E.M) of three independent experiments. The results of the average cytotoxicity at the highest concentrations tested are summarized in Table 10.

For foam extracts up to the test concentration of 40%, the cytotoxicity lower than 25% remained at a very low constant level. Then, the cytotoxicity depended on the foam type and increased from a test concentration of 40%. The strongest cytotoxicity was observed for foam extract labeled as VAB 3.

Figure 8 depicts morphological changes in the monolayer of HaCaT cells in the presence of polyurethane foam extracts. On the other hand, the ISO 10993-5 [28] standard’s qualitative morphological classification of cytotoxicity can be found in Table 11.

Keratinocytes (human epidermal cells) in the medium (negative control) had a regular, homogeneous shape with a well-visible cell membrane, cytoplasm, and nucleus. No cell lysis (decomposition) or restriction of their growth was observed, and a very small number of cells were detached from the monolayer and intracellular bodies (cell nuclei, mitochondria, etc.). After exposure to extracts produced from individual polyurethane foams, the cell monolayer was disrupted. Compared to the reference sample (negative control), the number of cells on the surface decreased—the microscopic analysis correlated with the results obtained from the cytotoxicity test.

A qualitative analysis of cell morphology was performed. A severe reaction of cells with the sample was observed on the positive DMSO sample. In the VAB 3 foam, the cell reaction was medium. The VAB 1 foam was mild, while no reaction was observed in the VAB 2 foam (Table 11).

## 4. Conclusions

This article describes polyurethane foams synthesized with three different sets of catalyst systems and verifies their effect on cytotoxicity and other properties of polyurethane foam. The catalyst system was the only variable in the formulation. The reactivity of the formulation containing the organometallic compound in the catalyst system was significantly higher than the rest of the systems, manifesting in significantly shorter synthesis parameters. FTIR analysis showed differences in the proportions of urethane and urea groups. The highest amount of urea groups characterized the foam with the organometallic compound in the composition. A slight effect of the catalysts can also be observed in the first stages of thermal degradation. A correlation was also observed between the SAG coefficient with the number of urea groups, which increased with an increased number of urea groups.

The morphological structures of the foams also differed, influencing the amount of sweat absorbed. A comparison of the bioassay results suggests that the best-performing foam toward skin cells was VAB 2, with a catalytic system consisting of commercial Diethanolamine and Addocat^®^105; however, it had lower mechanical properties. VAB 3, containing DBTDL, based on heavy metal, tin, occurring in large-scale manufacturing, had the biggest negative impact on skin cells. The foam formulas used in contact with the skin should avoid using these catalysts. Foam VAB 1, with a popular DABCO catalyst, had mild reactivity toward the cells, but we believe it should also be avoided. Despite the longer time of manufacture, the benefit of lesser cytotoxicity toward a human makes it a suitable dedication. Despite the lower characteristics of VAB 2 foam, the comfort value was satisfactory (above 1.9). All the polyurethane foams produced were characterized by better absorption of alkaline sweat, which is more favorable from the point of view of bacterial growth. When analyzing the results of the produced polyurethane foams, new correlations were observed between the chemical structure and properties obtained by using different catalytic systems. The differences in the properties of the foams presented in this article are relevant to their use as biomaterials remaining in contact with the skin for a short time. During future work, attention will be paid to improving the comfort factor of the VAB 2 foam while maintaining its biocompatibility so that it can be used safely in contact with the skin. It should also be checked if a higher SAG factor is more beneficial than the full biocompatibility of the foam but with a lower comfort factor.

## Figures and Tables

**Figure 1 materials-16-01527-f001:**
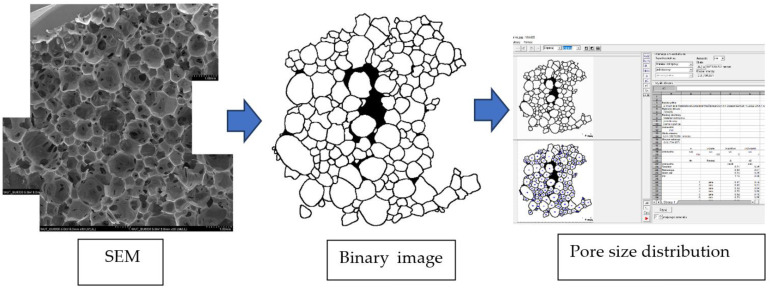
Research methodology applied in the course of the observation analysis (example on VAB 1 foam).

**Figure 2 materials-16-01527-f002:**
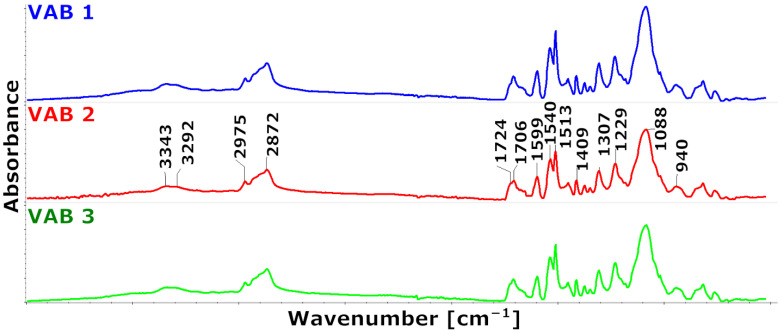
FTIR spectra of the produced foams.

**Figure 3 materials-16-01527-f003:**
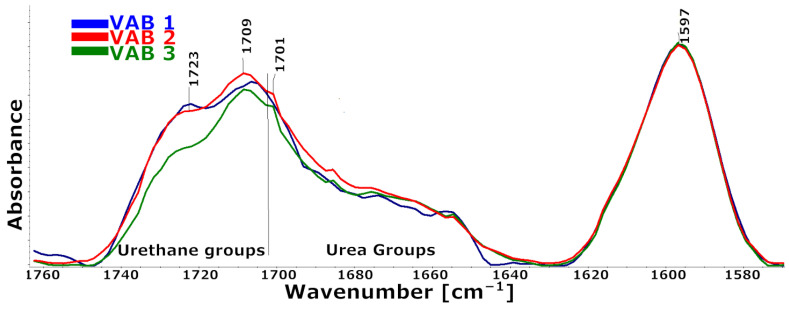
Fragment of the FTIR spectrum from the range of carbonyl groups 1630–1760 cm^−1^ and the band from the aromatic rings of the isocyanate used to scale the spectra.

**Figure 4 materials-16-01527-f004:**
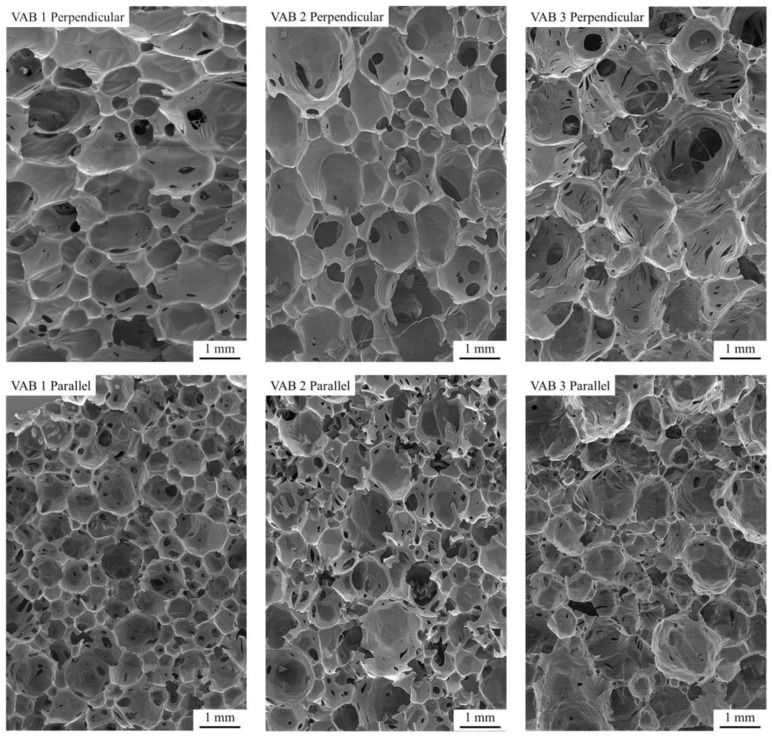
Images composed of SEM images of cross-sectional samples taken in perpendicular and parallel directions.

**Figure 5 materials-16-01527-f005:**
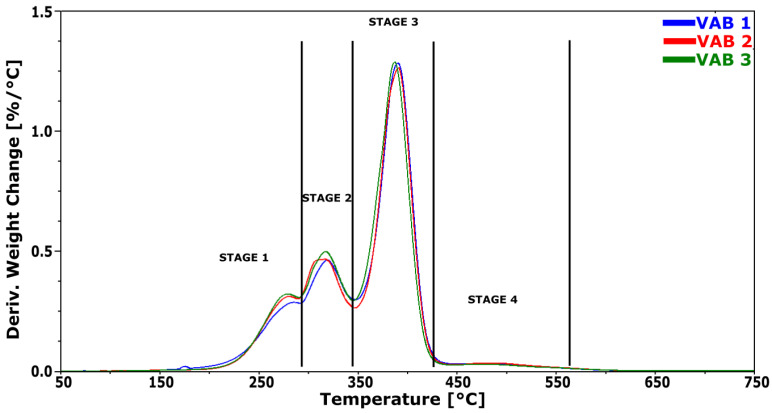
DTG thermograms obtained during TGA thermogravimetric analysis of foams.

**Figure 6 materials-16-01527-f006:**
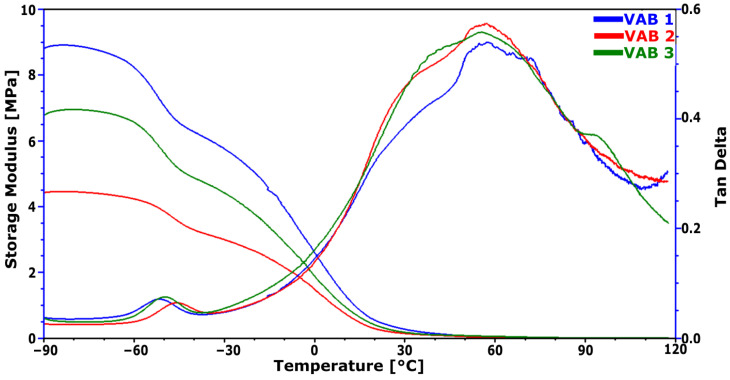
Results of DMA analysis of foams made from different catalyst systems.

**Figure 7 materials-16-01527-f007:**
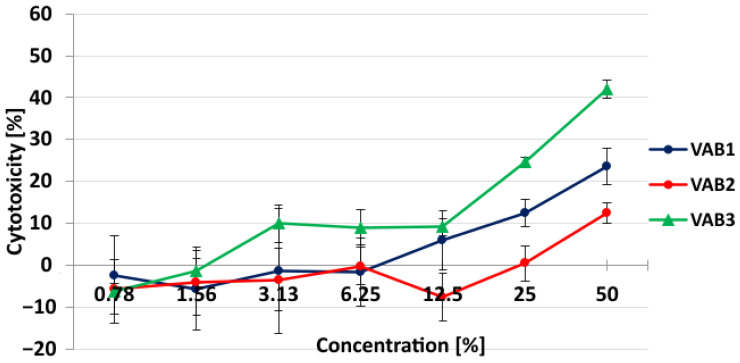
Cytotoxicity of polyurethane foam extracts after 24 h exposition of HaCaT cells (human keratinocyte) in Neutral Red Uptake assay. The standard deviation of the mean—S.E.M.—of the absorbance values of the four replicates from three distinct experiments is represented by each point.

**Figure 8 materials-16-01527-f008:**
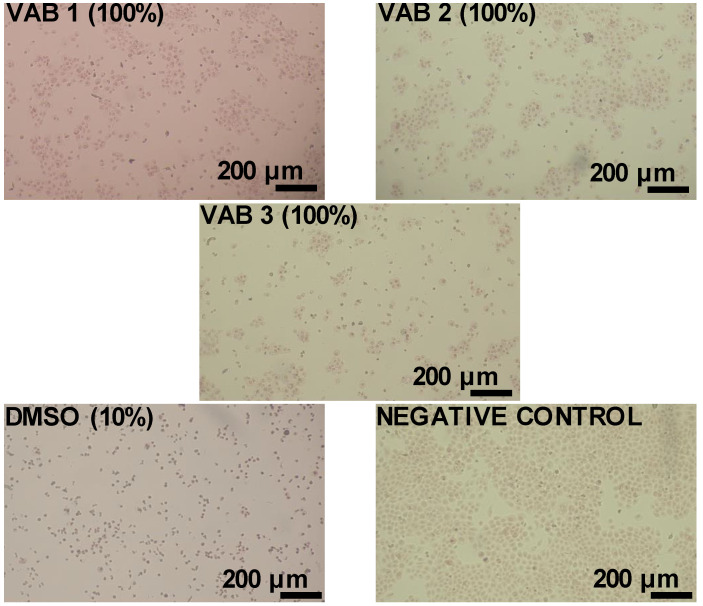
Microphotographs of human keratinocytes HaCaT after 24 h exposure to polyurethane foam extracts after neutral red. Objective 100× (Nikon Ts2, contrast EMBOSS, Tokyo, Japan).

**Table 1 materials-16-01527-t001:** Type of catalyst used in the produced polyurethane foams.

Sample	Catalysts
VAB 1	DABCO NE 1070, JEFFCAT^®^ DPA, JEFFCAT^®^ DMDPTA
VAB 2	DELA, Addocat^®^ 105
VAB 3	Niax A-1, Addocat^®^ 105, DBTDL

**Table 2 materials-16-01527-t002:** Foams synthesis parameters.

Sample	Parameter, s			Apparent Density, kg/m^3^
	Start Time	Rise Time	Gel Time	
VAB 1	17	420	560	44.91 ± 0.83
VAB 2	15	255	300	40.30 ± 0.68
VAB 3	2	60	90	40.00 ± 0.51

**Table 3 materials-16-01527-t003:** The results of the analysis of the DPS and the proportion of urethane bonds in the hard phase of foams.

Sample	DPS	Part of Urethane Groups, %	The Proportion of Hydrogen Bonds Linking Urethane Groups, %
VAB 1	1.31 ± 0.23	56.8 ± 1.8	57.6 ± 1.9
VAB 2	1.12 ± 0.12	62.7 ± 3.2	49.0 ± 1.7
VAB 3	1.61 ± 0.29	53.4 ± 1.5	53.9 ± 1.8

**Table 4 materials-16-01527-t004:** Pore sizes calculated on sections taken perpendicular and parallel to the direction of growth.

Sample	Mean Pore Diameter d_2_ (mm)		Coefficient of Variation of Mean Pore Diameter d_2_	
	Perpendicular	Parallel	Perpendicular	Parallel
VAB 1	0.55 ± 0.43	0.47 ± 0.25	0.78	0.54
VAB 2	0.41 ± 0.31	0.43 ± 0.22	0.75	0.51
VAB 3	0.63 ± 0.45	0.38 ± 0.33	0.71	0.86

**Table 5 materials-16-01527-t005:** Results of DTG curve analysis.

Sample Name	Stage 1			Stage 2			Stage 3			Stage 4		
	T_max1_	V_max1_	D_m1_, wt. %	T_max2_	V_max2_	D_m2_, wt. %	T_max2_	V_max2_	D_m3_, wt. %	T_max2_	V_max2_	D_m4_, wt. %
VAB 1	272.3	0.14	13.2	318.6	0.44	21.0	390.6	3.54	50.8	491.2	0.05	4.6
VAB 2	271.1	0.19	13.4	316.9	0.53	21.2	390.9	3.72	50.8	490.8	0.08	4.6
VAB 3	271.4	0.23	13.9	317.2	0.50	22.6	387.3	3.53	50.1	488.2	0.05	4.2

**Table 6 materials-16-01527-t006:** Results of elastic rebound and return time analysis.

Sample	Rebound Resilience, %	Elastic Recovery Time, s
VAB 1	11.17 ± 0.14	6.7
VAB 2	9.70 ± 0.19	6.0
VAB 3	12.67 ± 0.22	5.9

**Table 7 materials-16-01527-t007:** Results of the analysis of mechanical properties.

Sample	Compression Set		Hardness CV40%, Pa	SAG Factor
	50% (22 h, 70 °C), %	90% (22 h, 70 °C), %		
VAB 1	2	3	2.32	3.10
VAB 2	0	1	2.89	2.35
VAB 3	1	1	1.49	3.21

**Table 8 materials-16-01527-t008:** Results of the analysis of the curves obtained during the DMA analysis.

Sample Name	Tg, °C	E’-75 °C, MPa	E’100 °C, kPa	E’36 °C, MPa	Tan Delta 36 °C
VAB 1	−50.8	8.8	4.7	0.18	0.42
VAB 2	−45.9	4.4	4.4	0.11	0.49
VAB 3	−49.1	6.8	6.0	0.12	0.50

**Table 9 materials-16-01527-t009:** Results of the determination of water absorption by the foams.

	Sweat Uptake Coefficient gwet/gdry		Standard Deviation	
Sample Name	Base Sweat	Acid Sweat	Base Sweat	Acid Sweat
VAB 1	21.1	13.9	5.2	0.7
VAB 2	20.8	15.4	10.8	0.8
VAB 3	25.7	20.7	2.9	2.9

**Table 10 materials-16-01527-t010:** IC_50_ values of polyurethane foam extracts after 24 h exposition of HaCaT cells (human keratinocyte) and average cytotoxicity (%) at the highest tested concentration (±standard deviation of the mean—SEM).

Sample	Average Cytotoxicity (%) at the Highest Tested Concentration (±SEM)	Sequence of Cytotoxicity
Positive control (DMSO)	55.2 ± 8.3	The highest
VAB 1	23.6 ± 4.3	2
VAB 2	12.5 ± 2.5	3—the weakest
VAB 3	42.0 ± 2.2	1—the highest

**Table 11 materials-16-01527-t011:** Qualitative morphological grading of cytotoxicity of polyurethane foam extracts (100% concentrations) according to ISO 10993-5 observed in an inverted microscope before adding neutral red.

Sample	Grade	Reactivity	Conditions of All Cultures According to ISO 10993-5
Vehicle control	0	None	Discrete intracytoplasmic granules, no cell lysis, no reduction in cell growth.
Positive control (DMSO)	4	Severe	Nearly complete or complete destruction of the cell layer.
VAB 1	2	Mild	No more than 50% of the cells are round, devoid of intracytoplasmic granules; no extensive cell lysis; no more than 50% growth inhibition is observable.
VAB 2	0	None	Discrete intracytoplasmic granules, no cell lysis, no reduction in cell growth.
VAB 3	3	Moderate	More than 70% of the cell layers contain rounded cells or are lysed; cell layers are not completely destroyed, but more than 50% growth inhibition is observed.

## Data Availability

Experimental methods and results are available from the authors.

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
