# Peer review of "Effect of Different Amine Catalysts on the Thermomechanical and Cytotoxic Properties of ‘Visco’-Type Polyurethane Foam for Biomedical Applications"

_materials, 2023, doi:10.3390/ma16041527_

Round 1

Reviewer 1 Report

The paper shows how catalysts included in polyurethane composition affects some properties relevant for medical purposes. The topic is interesting and the paper well-written. Some changes are required before submission. 

1) From the first lines, Introduction is medicocentric. However, polyurethanes are also adopted in many other fields. Please, add some introduction lines in a general framework.  For example, in civil engineering recent examples are https://doi.org/10.1016/j.soildyn.2021.106602 , https://doi.org/10.21660/2021.82.Gx263  and https://doi.org/10.1007/s10518-022-01412-0  

2) Line 39-40: "Polyols, isocyanates..." the sentence appears to be incomplete. 

3)Table 1: Specify the meaning (if there is any) of VAB abbreviation 

4) Line 127 and 130: Please, replace 10x10x10 cm with 10x10x10 cm3.

5) Line 145-146: "After this time, the specimens were removed from the metal covers and measured again". Measured what? What property?

6) Figure 2 and 3 have absorbance in y-axis. Authors could consider to define what this quantity is in Section 2.2.2.

7) Line 347 and Line 351: Figure XXX and Table XXX. Replace XXX with the correct number. 

8) Line 351: What is tan delta?

9) Table 6: Is it correct the order of magnitude of the modulus changing from MPa to kPa,  depending on temperature? Or is kPa a mistake?

10) Table 7 is not mentioned in the Manuscript. 

11) Line 530: Readers could not know that PU is an abbreviation of Polyurethane. Please, introduce it before. 

12) The current Discussion section has the aspect of a Conclusion section. A proper Discussion section where results are compared with other literature works (if possible) should be included. 

Reviewer 2 Report

Dear Author/s
It is well written and interesting study focusing on dermatological modification in foam structure. I uploaded a file which has recommendations for the manuscript to put it in a better shape.
Best regards

Reviewer 3 Report

The authors reported the effect of amine catalysts on the thermomechanical and Cytotoxic Properties of ‘Visco’ Type Polyurethane Foam. The submission is unacceptable for the following points:-

 1.      The title should be revised to be short, precise, and informative. It should reflect the author’s main findings.

2.      The novelty of the study is low and should be improved.

3.      Please, explain the negative values of cytotoxicity shown in Figure 7.

4.      The thermal degradation products of the materials should be identified.

5.      The chemical composition of the foam should be fully investigated. It is important information for the biomedical study.

6.      The language should be revised and typos should be corrected.

Minors

 7.      Adjust subscript and superscript; for example, ‘(H2NCO)NH(CH2)3N(CH3)2’; ‘4000-400 cm-1.’;

8.      Correct units such as ‘10×10×10 cm’, it should be volume units.  Replce comma by period for ‘0,53 %/°C,’ 0,44 %/°C.’…..

Round 2

Reviewer 1 Report

Most of my suggestions have been addressed, except for the first one. The first sentence is now better but the Authors haven't included any mentions of the applications of earthquake engineering, as I had suggested, where a lot of effort has been done (Some examples: https://doi.org/10.1016/j.soildyn.2021.106602 , https://doi.org/10.21660/2021.82.Gx263 and https://doi.org/10.1007/s10518-022-01412-0).

Again, I recommend mentioning this. Once this is fulfilled, the Manuscript can be considered for publication.  

Author Response

Dear Reviewer, 

I have read the articles. I've decided to mention the articles about earthquake engineering in the first line. 

Best regards

Author Dominik Grzęda

Reviewer 3 Report

The authors addressed most of the comments and the revised version can be accepted.

Author Response

Dear Reviewer,

Thank you for your comment.

Best regards

Authors